# Ghana’s Adherence to PASCAR’s 10-Point Action Plan towards Hypertension Control: A Scoping Review

**DOI:** 10.3390/ijerph20021425

**Published:** 2023-01-12

**Authors:** Francis Sambah, Bunmi S. Malau-Aduli, Abdul-Aziz Seidu, Aduli E. O. Malau-Aduli, Theophilus I. Emeto

**Affiliations:** 1College of Public Health, Medical and Veterinary Sciences, James Cook University, Townsville, QLD 4811, Australia; 2Department of Sports and Exercise Science, College of Health and Allied Sciences, University of Cape Coast, Cape Coast P.O. Box UC 182, Ghana; 3College of Medicine and Dentistry, James Cook University, Townsville, QLD 4811, Australia; 4World Health Organization Collaborating Center for Vector-Borne and Neglected Tropical Diseases, James Cook University, Townsville, QLD 4811, Australia

**Keywords:** health policies, hypertension, Ghana, scoping review

## Abstract

The continuous increase in the prevalence of hypertension in Ghana has led to various interventions aimed at controlling the disease burden. Nonetheless, these interventions have yielded poor health outcomes. Subsequently, the Pan-African Society of Cardiology (PASCAR), established a 10-point action plan for inclusion in policies to aid control of hypertension. This scoping review assessed the adherence of health policies to the 10-point action plan towards hypertension control/reduction in Ghana. Eight health policies met the inclusion criteria and were assessed. The programme evaluation and policy design framework were used for synthesis and analysis of extracted data. Overall, there was poor adherence to hypertension control observed in the policies. Specifically, there were low levels of integrating hypertension control/reduction measures, a poor task-sharing approach, and poor financial resource allocations to tackle hypertension control/reduction in most of the policies. There was also low support for research to produce evidence to guide future interventions. For Ghana to achieve the global target of reducing hypertension by the year 2025, its health policies must adhere to evidence-based interventions in hypertension management/control. The study recommends a follow-up study among hypertension patients and healthcare professionals to evaluate the factors militating against hypertension management/control in Ghana.

## 1. Introduction

As reported by the World Health Organization (WHO), hypertension (HTN) is one of the most common chronic diseases [1]. It is also a major risk factor for many noncommunicable diseases (NCDs). It accounts for 43% of global disease burden (GDB) and 71% of yearly worldwide mortalities and disability-adjusted life years (DALYs) [2,3]. An estimated 12.8% of global mortalities are attributable to HTN [4]. Global epidemiological data indicate that 650 million persons were diagnosed with HTN as of 1990. By 2019, this figure had risen to 1.28 billion [1]. The prevalence of HTN is projected to increase by 30% by the year 2025 if ageing and population growth are not properly managed [5].

The prevalence of HTN is high in low-and-middle-income countries (LMIC) [6] and particularly in Africa, where it accounts for 11% of mortalities [1]. Bosu et al. [7] reported a pooled prevalence of 57% in Africa, an increase over a previous report of 48% in 2016 [8]. This prevalence is disproportionate across nations within Africa. For example, HTN prevalence is reported to range from 27–58% in South Africa [9,10], 38.1% in Nigeria [11], and 24.5% in Kenya [12]. Even more disturbing is the increasing prevalence of HTN among rural residents hampered by high resource inequalities ranging from healthcare accessibility, economic resources, human (working population), and poor nutrition [13,14].

In Ghana, like many other African countries, the prevalence of HTN is relatively high at 30.1% [15]. HTN affects one in four persons in Ghana, accounting for 4.7% of overall hospital admissions and 15.3% of associated deaths [16]. Furthermore, the disease is no longer restricted to the wealthy urban elite but is increasingly endemic in the rural populace [17]. This suggests that the problem has now spread across the Ghanaian society, with a reported rural-urban prevalence of 5.1% vs. 9.5% [17].

The economic impact of HTN in Africa cannot be underestimated, as the cost of care associated with HTN risk factors and comorbidities, such as stroke, ischemic heart disease, and congestive heart failure, continues to be the leading cause of considerable financial hardship [18]. Direct healthcare expenditures associated with treating HTN and its risk factors are a considerable financial burden. In addition, there are associated secondary costs such as lost savings and assets when families are forced to pay for unplanned healthcare expenses, including rehabilitation after a stroke or dialysis after kidney failure [19]. There are also significant financial and social opportunity costs for families who, in the absence of institutional care systems, must offer intensive long-term care to aging relatives with HTN [19]. These economic burdens of HTN necessitate the consideration of non-pharmacological interventions proven to be efficacious in the control and management of HTN.

Tackling chronic conditions such as HTN requires a comprehensive, broad base, nationally oriented implementation plan and documentation that galvanizes stakeholders, interest groups, and resources to achieve. Various health policies have been implemented in the past; nonetheless, these policies have yielded minimal control of NCDs, especially HTN. Given the overwhelming levels of HTN prevalence (27% globally; [20], 25.9% in sub-Saharan Africa (SSA) [21], and 30.1% in Ghana [22]), African governments noted through evidence-based research that previously implemented interventions had poor health outcomes [23]. Subsequently, the African governments, in a consensus statement through the Pan-African Society of Cardiology (PASCAR), established a 10-point action plan to help in addressing the menace of HTN in the continent and tasked individual countries to adopt and incorporate the 10-point action plan into health policy formulations and strategic frameworks with the aim of achieving 25% HTN reduction by 2025 [23]. This 10-point action plan recommends that all NCD national programs include a plan for HTN detection and allocate appropriate funding and resources for the early detection, efficient treatment, and control of the disease. It further recommended the creation or adoption of simple and practical clinical evidence based HTN management guidelines supplemented by annual monitoring and reports of the detection, treatment, and control rates of HTN, with a clear target of improvement by 2025. The goal is to facilitate a task-sharing approach with adequately trained community health workers and ensure the availability of essential equipment and medicines for managing HTN at all levels of care.

In response to the PASCAR consensus statement, Kenya [24] and Ghana [25] have published NCD policies aiming to achieve the 2025 target of 25% reduction or control of HTN. A review of prior policies implemented by Ghana and Kenya highlighted some flaws that need to be eschewed in the current policies if achieving the 2025 target will be practically plausible. For example, the August 2012 NCD policy implemented by Ghana aimed to lower the incidence, prevalence, and exposure of people to NCD risk as well as reduce NCD-related morbidity and improve the overall quality of life of people with NCDs [26]. It focused on the use of primary preventive and clinical care, such as the early identification and treatment of HTN, health system strengthening including health worker training, and human resource capacity building [26]. However, this document was flawed with a lack of impactful indicators for each of the NCDs to enable the tracking of progress and lumped HTN with CVDs, which could mask strategies targeted at addressing HTN specifically and evaluating progress appropriately. Similar findings were noted in the previous NCD policy implemented by Kenya. If these bottlenecks are not addressed and curtailed in the recently published policy frameworks across Africa, achieving the 25% HTN reduction/control target by 2025 with appropriate health indicators may be impossible.

Reviews conducted so far on HTN in Ghana have focused on its prevalence and associated factors [15,27] as well as impact of physical activity [28]. These limited reviews failed to adduce evidence on Ghana’s policies response towards HTN control. This scoping review was conducted as a barometric assessment of the country’s preparedness towards achieving the 2025 target. The review assessed gaps in Ghana’s policies relative to the 10-point action plan advocated by the African governments for consideration in policy formulations to achieve the target of 25% reduction in the prevalence of HTN by 2025.

## 2. Materials and Methods

The review followed the five primary steps of Arksey and O’Malley’s [29] methodological approach for scoping review: (1) “Defining the research questions”, (2) “finding relevant studies (policies/documents)”, (3) “study selection”, (4) “data charting”, and (5) “collating, summarising, and reporting the findings”. This methodological framework has been employed in previous studies [30,31,32]. The study also followed a guideline for Preferred Reporting Items for Systematic Reviews and Meta-Analyses extension for Scoping Reviews (PRISMA-ScR) (see Appendix A).

### 2.1. Stage 1: The Review Questions

The following questions guided the review process:What are the characteristics of health policies developed from 2017 to 2022 for the control/reduction of hypertension in Ghana?To what extent do the current health policies adhere to the 10-point action plan for hypertension control/reduction in Ghana?

To guide the analysis, a conceptual framework for evaluating programme and policy design [33] was adapted (Figure 1). This framework has been used to evaluate programmes and policy designs in other studies [30,31,33]. The framework was adopted, as it enabled the assessment of the early stages of policymaking processes: political recognition, policy formulation, planned implementation, and monitoring and evaluation plan [33]. It is also applicable in assessing implementation factors such as the level of coordination, collaboration, and stakeholder engagement in the development of policy and programme design and in monitoring adherence to certain policy recommendations for the control and management of HTN [33]. In adapting the framework for this study, political recognition refers to the extent to which HTN control issues are recognised as national issues and prioritised. This may be seen in political leaders’ statements expressing their support for HTN prevention and control. The formulation aspect also considers the policy’s year of creation and state of development as well as whether the policy is new or a reorientation of existing policy and whether it is a national/international or governmental/nongovernmental initiative, the target group’s definition, the HTN control guidelines addressed, and policy objectives and strategies for achieving stated goals. The scope of actions and the financial and human resources used to reduce/control HTN prevalence among exposed and unexposed persons are all considered in the implementation plan. The monitoring and evaluation (M&E) plan centred on the monitoring methods and whether the policies have evaluation strategy. The roles as well as the number of partners involved in public and private activities are investigated in terms of level of coordination and collaboration, with the amount of coordination among policy players viewed as an indicator of likely success. Moreover, the main central focus of the review was to evaluate the policies’ adherence to PASCARs 10-point action plan for the control and management of HTN through policy initiation. Finally, at each stage of the policy formulation process, the nature of involvement of healthcare practitioners and HTN patients was considered [30,31,33]. 

### 2.2. Stage 2: Identifying Relevant Policies

Ghanaian health policy documents were used in this review. The inclusion criteria included (a) health policy documents enacted from 2017 to 2022, (b) policy available online, and (c) policy targets people aged 18 years and above. Policies excluded were those (a) published before 2017, (b) unavailable online, and (c) non-health related.

Databases such as *Scopus*, *Medline/PubMed*, *CINAHL/EBSCO*, and *Web of Science* were the starting point for the initial search. Search terms specifying “interventions on NCDs” OR “hypertension” OR “raised blood pressure” OR “elevated blood pressure awareness” OR “education” OR “promotion” were employed. Other search terms and phrases included “NCDs policy” OR “strategic plan of Ghana” OR “policy response to HTN” OR “CVDs control” OR “NCDs and healthcare providers capacity building”. Other sources such as *Google Scholar*, the Ministries of Ghana websites (https://www.ghana.gov.gh/search/?filterBy=ministry), ResearchGate, the Ghana Health Service websites (https://www.moh.gov.gh/policy-documents/) were searched.

### 2.3. Stage 3: Study Selection

The PRISMA-ScR flow diagram [34] presented in Figure 2, illustrates the screening technique utilised to identify the relevant policy documents for this review.

### 2.4. Stage 4: Charting the Data

Microsoft excel was used to create a matrix for extracting data from the policies (see Table 1 and Table 2). The ten constructs that guided the review include (1) “political recognition”, (2) “policy initiation”, (3) “target group definition”, (4) “policy objectives”, (5) “scope of activities”, (6) “financial and human resources”, (7) “monitoring and evaluation plan” (8) “level of cooperation and collaboration”, and (10) “level of health professionals and HTN patients’ involvement in health policies/reports in Ghana”. Further data were extracted to reflect PASCAR’s 10-point action plan for HTN control as shown in Table 2.

### 2.5. Stage 5: Collating, Summarising, and Reporting the Results

The policies were assessed for adherence to the PASCAR’s 10-point action plan using the conceptual framework (Figure 1). This assessment involved a systematic coding and classifying approach for analysing huge volumes of textual data to uncover trends/patterns in the words used, their frequency, linkages, and communication structures and discourses [35,36].

## 3. Results

Eight health policies met the inclusion criteria (Table 1 and Table 2): the Non-Communicable Disease (NCDs) Policy [25], the National Health Policy (NHP) [37], the Universal Health Care Policy (UHC) [38], the Health Commodity Supply Chain Master Plan (HCSCMP) [39], the Health Sector Medium-term Development Plan (HSMTD) [40], the National Medicines Policy (NMP) [41], the National Food Safety Policy (NFSP) [42], and the Ministry of Health Client Service Charter (MHCSC) [43]. The results are divided into two parts. Part one describes the assessment of the included policies based on the adapted conceptual framework for evaluating programme and policy design [32], and part two assesses the included policies on their adherence to PASCAR’s 10-point action plan for hypertension control.

### 3.1. Part One: Evaluation of Programme and Policy Design of Included Studies

Table 1 depicts the characteristics of the health policies/reports extracted based on the framework (Figure 1). These key points are highlighted below.

#### 3.1.1. Political Recognition

All the eight policies [25,37,38,39,40,41,42,43] demonstrated evidence of political endorsement. The President or, in some situations, the designated relevant minister signed the policy document, indicating their political recognition. For example, the President of the Republic of Ghana and the minister of health both signed the NHP [37], while only the minister of health signed the Ghana UHC [38]. The NCDs policy was signed by the President of Ghana, the minister of health, and the minister of finance, indicating political commitment and recognition of these policies [24].

#### 3.1.2. Policy Initiation

Selected policies were initiated from 2017 to 2022. The NHP (37) and NCDs policies [24] are revised policies, whilst the UHC policy is a new policy [38]. All included policies had a multifaceted approach to health, and their overarching goal is to improve health outcomes by making healthcare accessible for all [38], equitable, and affordable irrespective of age and social class [41]. They also tackle risk factors that affect cardiovascular health, such as HTN [24].

#### 3.1.3. Target Group Definition

The target populations covered by these polices/reports were mixed. While some measures were directed at specific groups, the majority were directed at the general public. The NCDs policy, for example, is aimed at both the public and persons living with NCDs such as HTN (24). The other policies included NCDs or cardiovascular disorders such as HTN [38,40]. However, neither NCDs nor HTN were mentioned in the HCSCMP [39], NFSP [42], and MHCSC [43].

#### 3.1.4. Composition of the People Who Developed the Policy/Advisory Panel

The composition of the personnel who formulated the various policies enacted was another important element in this review. While the majority of the policies (e.g., [25,37,38,41,42] said that they engaged a variety of agencies such as the West African Health Organisation, UNICEF, Ghana College of Pharmacists, and Nursing and Midwifery Council and medical practitioners such as Dr. Anarfi Asamoah Baah (Former Deputy Director General, WHO), Dr. Patrick Aboagye (Director General, Ghana Health Service), and Dr. Abena-Tannor (Komfo Anokye Teaching Hospital), when crafting the policies, there was no particular mention of groups affiliated with HTN.

#### 3.1.5. Hypertension Issues Discussed

This component of the evaluation concentrated on HTN-related issues raised in the policies. While some of the policies made no mention of HTN [39,42,43], others had sections dedicated to the subject [25,37,38,40,41]. Five policies addressed the management and control of HTN [25,37,38,40,41]. The NHP addressed HTN-related concerns such as offering assistance and empowerment to the public to encourage the adoption of a healthier lifestyle [37]. The UHC policy highlighted the need to optimise critical preventive healthcare services that include HTN.

The NCDs policy covers general NCDs-related issues without explicitly detailing HTN-specific control measures. Meanwhile, HTN is the major risk factor for many NCD conditions and should be prioritised since the control of HTN may directly lead to the management and control of many NCDs [2,3].

#### 3.1.6. Policy Objectives

All eight policies described their objectives in general and specific terms. These objectives were also stated qualitatively; for example, the NMP in one of its objectives states the following:

*“To ensure that medicines selected for incorporation in the Essential Medicines List are suitable for the appropriate treatment of prevailing diseases, and that people’s medicines needs at different levels of the health care system are met in the most scientifically sound and cost-effective manner”* [41] (pp. 14, 53).

Despite the lack of strategies and measurability of the policy objectives, the document however, communicated an overall quantitative measurable goal with indicator targets that state the following:

*“…to attain at least 80% coverage of Ghanaians having access to essential health services”, “Reduce by one-third pre-mature mortality from non-communicable diseases…”* [37] (p. 19).

In the main NCDs policy document, the policy objectives were similarly qualitative, without measurable targets and indicators [25] (pp. 16,23–24).

#### 3.1.7. Scope of Activities

All included policies indicated a range of activities and modalities to help attain policy objectives. For example, the policies sought to use advocacy, education, health promotion, empowerment, behaviour change communication, access provision, skill training, capacity building, institutionalisation, decentralisation, research communication, legislation enforcement, machinery procurement, and essential drugs supply to achieve policy aims [25,36,37].

#### 3.1.8. Financial and Human Resources

To design policies and effectively implement them, both human (that is, the skills needed to develop and execute the policy) and financial resources (the monetary incentives) are required. The human resources include educators, health promotors, disease surveillance officers, ministries, development partners, non-governmental organisations (NGOs), academia, and research institutions. All eight policies mentioned government and donor agencies such as the West African Health Organisation, Centre for Plant Medicine Research, Japan International Cooperation Agency, Ministry of Health, and many others [25,37,41]. Few of the policies have detailed budgetary allocations to help implement the stated objectives with stipulated funding sources [40,41]. Others have no stipulated funding/budget [42,43]. Regarding human resources, the NCDs policy captured a clear vision on the needed human resource plan for the policy implementation and success. The following excerpt buttresses the policy human resource plan:

*“…efforts will be made to produce and equitably distribute motivated human resource needed for NCDs prevention and control. Capacity building of staff that exists under the various levels will include the application of task shifting and task strengthening strategies”* [25] (p. 30).

#### 3.1.9. Monitoring and Evaluation (M&E) Plan

The study also sought to assess whether the policies included an M&E strategy and performance indicators and, if they did, whether the indicators were realistic and relevant to the policies’ goals. Although all the eight policies discussed M&E plans, few included a developed M&E plan that reflected policy objectives and indicator targets. For instance, the NFSP [41] states the following:

*“…monitoring and evaluation of the policy will be the responsibility of the Ministry of Health, in collaboration with the Food and Drugs Authority and the National Food Safety Intersectoral Committee. As part of the process an effective M&E system will be built into the strategic framework from the onset. The system will monitor programme implementation and performance against a set of pre-determined indicators at all levels”* [41] (p.31).

However, the same policy includes no indication of any realistic framework, indicator targets, and baseline for comparison. In contrast, the NHP document had an elaborate M&E plan with outcome indicator targets incorporated as well as timeframes for M&E as stated:

*“Monitoring of the progress and achievement of the health outcomes will be routine and continuous (quarterly, half-yearly and annual)”* [37] (pp. 34–35,37).

Moreover, the policy on UHC [38] captured an M&E section with indicator targets and baseline for comparison. It shows both qualitative and quantitative estimates of objective targets to be achieved.

Lastly, in the NCDs policy, though it recognises the need for a formal evaluation of the policy impact in controlling NCDs, the mode of evaluation, targets indicators, timelines, and baseline measurements are not indicated [25] (pp. 31–32). Additionally, the method and type of evaluation (i.e., process or impact) to be employed is not shown.

#### 3.1.10. Level of Coordination and Collaboration

This study also included information about the extent of cooperation and coordination on various policies. Most of the policies were developed in collaboration with multi-national and NGOs such as United States Agency for International Development (USAID), the World Food Programme (WFP), the European Union, WHO, and United Nations Children’s Fund (UNICEF), The World Bank, and Food and Agricultural Organisation (FAO). For example, the World Bank and the Japan International Cooperation Agency provided financial and technical support for the development of the UHC policy [38]. The need for coordination and collaboration was recognised in all included policies.

#### 3.1.11. Level of HTN Patients Involvement

At each stage of the policy design process, from formulation through the implementation plan, monitoring, and evaluation, the extent of HTN patients’ or organisations’ involvement was investigated. The direct and indirect levels of involvement was differentiated. Surveys with HTN patients or organisations, HTN focus groups, and informal feedback from HTN patients in the field are examples of indirect engagement. Direct involvement refers to initiatives in which HTN patients and networks participate as collaborators in policymaking. The types of data utilised to establish the policies implied that HTN patients and their networks were involved in surveys, focus groups, and other data collection methods. However, there is little evidence in the policies of direct involvement of HTN patients and their networks during and through the policies preparations. Further, few data in two [25,38] of the reviewed policies show some indirect involvement of HTN patients or their networks. For example, an excerpt from the UHC policy states as the following:

*“There is observed increase in noncommunicable diseases particularly for hypertension…among the general population”* [38] (p. 4).

For the NCDs policy, the involvement of civil society organisations, which is not specific, may be inferred as involvement of HTN interest-group people [25] (p. 15). Lastly, the NHP had evidence of indirect involvement of HTN patients or probable organisations, as it shows evidence of the rising burden of HTN. For example, the policy document expresses the following:

*“NCDs such as hypertension…and related conditions are increasing in prevalence”* [37] (p. 13).

These indirect engagements of HTN patients by the policy formulators are not sufficient to adequately address concerns of the disease to enhance control and management.

### 3.2. Part Two: Evaluation of Included Policies on the PASCAR Guidelines

Table 2 shows the level of adherence to the 10-point action plan towards HTN control contained in the included policies. The key points are described below.

#### 3.2.1. Population-Level Interventions for Preventing Hypertension

This component of the evaluation examined interventions at the population level, aimed at controlling and managing HTN as captured in the policy documents. All eight policies stipulated actions from the individual and the community levels to help enhance population health. Though not all the policies specifically targeted their interventions at HTN, their interventions have direct and indirect positive effects on HTN management and control. For example, the NHP [37] aims to empower the population to adopt healthy lifestyles, which may reduce the risk factors associated with HTN. The following excerpts from the policy document support this assertion:

*“The population will be empowered and supported to proactively take measures to adopt healthy lifestyles. The policy shall ensure that the individual is encouraged to enjoy and enjoys adequate rest and shall support the development of recreational and physical activity facilities for regular use of the population towards the achievement of long-term individual and population health benefits. In addition, abstinence from alcohol or moderation in alcohol consumption will be encouraged and promoted through the strengthening of regulations on the production, marketing, and sale of alcoholic beverages”* [37] (pp. 23–25).

Further, the UHC policy [37], in its list of essential services for Ghanaians, indicated optimising basic essential universal services from primary healthcare levels, the prevention of chronic health conditions, and the promotion of services targeted at critical health conditions including HTN. Excerpts from the policy interventions include the following:

*“… promotive services, control of use of alcohol, tobacco and harmful substances; awareness on: regular medical check-ups…healthy eating, physical activity and wellbeing…”* [37] (p. 6).

#### 3.2.2. Additional Plan for the Detection of Hypertension

This section seeks to identify policy initiatives that incorporated screening for early warning signs of HTN, which include the risk factors for the manifestation of the condition. Only two of the policies reviewed [24,38] stipulated initiatives for the early detection of HTN. The others did not have plans for the detection of HTN.

Furthermore, the NCDs policy launched various modalities to be adopted and implemented, ranging from risk factors education, NCDs screening services, and early warning detection signs. These are intended to be implemented and linked to care for the control of NCDs including HTN. The following is an excerpt from the policy:

*“…early detection targets persons with NCDs symptoms and persons with no NCDs symptoms but who are at risk of NCDs. This policy seeks to establish screening services to contribute to the reduction of NCDs morbidity and mortality through the implementation of national guidelines for screening, the establishment of wellness clinics and the strengthening of the capacity of the community-based Health Planning and Services (CHPS) to provide screening services”* [25] (p. 19).

#### 3.2.3. Funding and Resources for the Early Detection, Efficient Treatment, and Control of Hypertension

For the success of any formulated policy, financial and human resources are pivotal. These resources are required from the developmental stages all through to implementation and evaluation. The human resources required for the execution of policies include skilled personnel such as healthcare professionals, specialists, disease control and surveillance personnel, researchers, developmental partners, and health promoters, while monetary incentives are required for the procurement of essentials. For these eight policies reviewed, only one policy highlighted human and financial resources to enable the detection, treatment, and control of NCDs including HTN [25]. The others either mentioned financial and human resources in general but not specifically on HTN management and control. In the NCDs policy document, several portions explore both human and financial arrangements aimed at enabling the successful implementation of the policy. For example, there were mentions of prioritising the production of healthcare professionals who are in short supply and the equitable distribution of health professionals to enable all regions to establish NCDs clinics even at the CHPS level [25] (p. 31). Excerpts from the document are given below:

*“…in line with the Government’s “Ghana beyond aid” agenda there is the need to explore innovative ways for funding NCDs prevention and control activities. This will be done in line with domestic, bilateral and multilateral funding mechanisms and within the current health financing arrangements. The policy will advocate for earmarked funds and increased budgetary allocations for NCDs”* [25] (p. 31).

However, specific resource allocations are not indicated in the policy document, leaving room for discretions that may affect critical NCDs such as HTN.

#### 3.2.4. Adopt Simple and Practical Clinical Evidence-Based Hypertension Management Guidelines

The treatment of HTN requires evidence-based guidelines that serve as the standard guiding protocol for the detection, treatment, management, and control of HTN. Nations, therefore, adopt these standard guidelines and incorporate them in their health policies for the administration of care and treatment for disease conditions. The HTN guidelines contain clinical evidence-based protocol such as the establishment of blood pressure threshold for the start of HTN medication. It establishes whether laboratory testing or a cardiovascular risk assessment are required before starting HTN treatment. It guides the selection of the pharmacological agents with which to begin treatment and helps determine whether to begin treatment with monotherapy, dual therapy, or single-pill combinations. It establishes blood pressure control targets in hypertension; follow-up intervals for HTN patients; and determines how nonphysician health care workers can help with HTN management [40,41]. All included policies failed to mention any clinical guidelines or treatment protocol for HTN. However, the NCDs policy and the NMP document highlighted a standard treatment guideline that is not specific to any NCD [25,41].

#### 3.2.5. Support High-Quality Research to Produce Evidence That will Guide Interventions

The role of research in healthcare is non-negotiable, as it offers new evidence for the continuity of care, improvement of care services, and provision of the right mix of intervention and/or treatment guideline for health. This review explored whether the policies created an opportunity for research and how the research will be supported. Most (5 out of 8) [25,36,37,39,42] of the included policies provided an opportunity for research. However, how and when the research will be conducted and by who as well as from what funding sources is unclear in some of the policy documents [25,42]. For instance, the NHP contained a research agenda that will provide evidence for policy options and new knowledge, but allocated funding to conduct research was lacking [37]. However, both [38,41] had a clear path for research and provided a funding support plan, as evident in the following excerpt:

*“A national health research agenda will also be developed and funded by various partners. Academia, expert consultant, and research institutions will play a key role in its implementation”* [38] (p. 36).

Furthermore, the NCDs policy outlined a clear research and development section with clear research vision, but the type of research and funding targeted was not indicated [25] (p. 31).

#### 3.2.6. Monitoring and Evaluation

The study investigated whether the policies included a monitoring and evaluation plan and performance indicators and, if so, if the indicators were realistic and relevant to the policies’ objectives. Only four [25,37,40,41] of the policies examined included monitoring and evaluation plans that were fully constructed and represented policy objectives and indicator targets. The NHP, the NCDs policy [25], the UHC policies [38], and the HSMTDP [40] all made provisions for monitoring and evaluation. However, only the NHP [37] and the HSMTDP [40] documents had a constructed monitoring and evaluation framework with both quantitative and qualitative estimated outcome targets, which is reflective of policy objectives for monitoring and evaluation [37] (pp. 34–35). Further, the NCDs policy had an incomplete monitoring and evaluation plan where indicator levels and targets were not indicated [25] (p. 32).

#### 3.2.7. Integrate Hypertension Detection, Treatment, and Control within Existing Health Services

Controlling and managing HTN cannot be achieved in isolation of other health conditions, considering that resources for healthcare delivery services are limited in Ghana. Therefore, an integrated disease surveillance and management system would lead to better resource (i.e., both human and financial) utilisation, where more resource conditions (e.g., HIV/AIDS, tuberculosis, and maternal and child health) are utilised for the benefit of other less-endowed conditions such as HTN. As can be seen in the policy documents, seven out of eight of the policies, namely the NHP, the UHC policy, HCSCMP, the HSMTDP, the NMP, the NFSP, and the MHCSC, had no integration plan or aim for the detection and control of HTN. Only the NCDs policy had indications of integration to optimise resource management and disease control [25]. An excerpt from the policy is given below:

*“Integrate NCDs policy into all sector planning, budgeting and financial management systems for efficiency and sustainability of services”* [25] (p. 24).

#### 3.2.8. Promote a Task-Sharing Approach with Adequately Trained Community Health Workers

Almost all (7 out of 8) of the policies [37,38,39,40,41,42,43] did not have a task sharing/shifting approach. Only the NCDs policy had a task-shifting or sharing approach [25] (p. 30). However, almost all the policies acknowledged in one way or the other the need to build staff capacity, in-service training and training on the job, and collaborating with academic institutions to mount short courses to upgrade professional skills for the successful implementation of their policies.

#### 3.2.9. Ensure Availability of Essential Equipment and Medicines for Managing Hypertension at All Levels of Care

The management and control of HTN requires the availability of screening equipment such as oscillometric (automated) blood pressure devices, ambulatory blood pressure monitors (ABPM), or sphygmomanometers and antihypertensive medications. From the review, four policies/reports captured provisions for HTN screening equipment and strengthening local production of quality medications [25,37,38,41]. The other four policies did not incorporate issues of equipment and medication [39,40,42,43]. For example, the NHP expresses its commitment to improve technological equipment infrastructure as well as enhance the production and procurement of affordable quality medication that will offer assistive services for quality health delivery. It also envisioned to eradicate and streamline traditional medicine within the health services, as the following excerpt from the policy document shows:

*“Government will focus on the modernization/re-tooling of existing facilities, rationalize the construction and siting of additional purpose-built facilities, promote the availability and use of high-quality assistive devices and technologies (including prostheses, orthoses etc.) at an affordable cost. Comprehensive Health Technology Assessments (HTA) will be institutionalized and inform the selection and procurement of all medical technologies required. In addition, the availability, affordability, efficacy and overall quality of medicines and medical products for all recognized forms of medical practice, will be pursued”* [37] (pp. 20–21).

Similarly, the UHC policy outlined strategies to reform the supply chain management system of essential drugs, which includes antihypertensive medications, as well as to provide the enabling logistics that may help track supply and stocks:

*“All primary health care levels will be re-stocked with essential tracer drugs equivalent to three months of their medicine and non-drug consumables requirement. This will also serve as a re-capitalization process following years of indebtedness and stockouts. The aim is to improve the visibility of stocks and consumption at facility level”* [38] (pp. 11–12).

For the NCDs policy, the focus is to enhance access to essential medications, diagnostic equipment, and quality supply of essential medications for NCDs [25] (p.30). Several excerpts from the policy document are as follows:

*“This policy seeks to improve access to essential medicines and supplies for NCDs through the implementation of clearly outlined strategies”* [25] (p. 30).

*Further, the NMP also envisioned to ensure the availability of essential medicines at affordable and accessible to all Ghanaians [39]. An excerpt from the policy document states, “Government shall exempt selected essential medicines from Value Added Tax (VAT) and other forms of taxation. Such exempted drugs shall be reviewed periodically, but not beyond two years”* [39] (p. 24).

*“To improve the medicines pricing governance mechanisms and promote affordability of medicines in Ghana”* [39] (p. 22).

#### 3.2.10. Provide Universal Access and Coverage for Detecting, Treating, and Controlling HTN

Three of the reviewed policies (NHP, UHC, and NCD) had provisions to provide the public with universal access to HTN screening and treatment services across all levels of healthcare [25,37,38]. For example, the NCDs policy aims at primary and secondary preventions by advocating for screening for all Ghanaians, as expressed below:

*“…promotion of prevention and routine screening for diagnoses for NCDs. Increase access to care for NCDs at all levels. Health education in schools on all NCDs”* [25] (p. 19, 23).

Furthermore, the UHC policy recognises the need to broaden coverage in disease detection and control, including HTN, and thus captured it in its essential services to be optimised for the population health. For example, it articulates the following:

*“This roadmap recognizes the importance of all services. It however places greater emphasis on interventions that needed to be consolidated, scaled up and to attain universal health coverage”* [37] (p. 6).

This is captured in the policy as universal healthcare services that include primary, preventive, promotive, and rehabilitation services, which must be available to the populace. In the case of the NHP, it emphasises the following:

*“The healthcare delivery system will be strengthened to achieve Universal Health Coverage meaning: all persons living in Ghana will have the opportunity to access quality healthcare services they require, wherever they are, with cost of care not being a barrier”* [37] (p. 18).

Further, it states the following:

*“The policy will seek to strengthen surveillance and response systems to prevent, detect, investigate, protect against, control, and provide a public health response to the spread of diseases….” “The system shall operate across all levels (community, district, metropolitan, regional, and national) and recognize the animal-human interface as well (one-health)”* [37] (p. 18).

The other policies [42,43] did not have clear modalities for universal health coverage for the detection, control, and treatment of HTN.

## 4. Discussion

This scoping review assessed Ghana’s adherence to PASCAR’s 10-point action plan for HTN control in Africa. Eight policies [25,37,38,39,40,41,42,43] were reviewed. The findings show Ghana’s policies adherence to PASCAR’s 10-point action plan towards HTN control/reduction was low. Specifically, most policies had low levels of integration for HTN control [39,40,41,42,43] and poor task-sharing approach [37,38,39,40,41,42,43] due to a shortage of specialist health workforce in HTN management. Moreover, many of the policies showed poor adoption of practical, evidence-based HTN management and control guidelines [37,38,39,40,41,42,43] as well as low support for research to produce evidence to guide interventions [38,40,41,42]. Notwithstanding, a few policies included fair provision for universal coverage for HTN detection, treatment, and control [24,37,38] as well as provisions for available essential equipment and medicines for HTN management and control/reduction [25,38,41]. However, all the policies had population-level interventions for HTN management [25,37,38,39,40,41,42,43].

The study findings show that although Ghana has some health policies targeted at improving population health, the policies’ general adherence to PASCAR’s 10-point action plan for the control/reduction of HTN was low. Hence, Ghana’s ability to meet the WHO target of 25% reduction/control of HTN by 2025 could be far from sight, implying that many at-risk Ghanaians would continue to experience the HTN unless its control/reduction interventions are urgently implemented across the continuum of care. This study therefore recommends that stakeholders within the Ghana Health Service should consider pharmacological and non-pharmacological interventions for the exposed and at-risk populations, respectively.

This study also revealed that although the policies generally had wider political recognition, the involvement of HTN patients in the policies development and implementation processes was not clearly stated. This concurs with the finding by Owusu [44] that not all stakeholders participated in the NCDs policy formulation process. Similar findings were reported in Kenya that NCDs patients and their groups had low involvement in the first strategic NCDs policy process that was implemented [24]. This void may create apathy, poor awareness and advocacy, and lack of support and collaboration from HTN patients. This has been buttressed in previous evidence that when key stakeholders who are intended beneficiaries in a policy product are not involved in the policy development process, it could lead to poor support in the implementation process [45,46].

Comparing the findings of the present study with similar health policies across SSA, some discrepancies and similarities are noted. For example, similar policies in Kenya [24], South Africa [47], and Nigeria [48] highlight the involvement and participation of NCD alliance/community-based organisations (CBOs), which may include HTN patients in the NCDs policy processes. In South Africa, a multi-sectoral action involving a wide range of actors from sub-districts at the provincial to the national levels participated in the various stages of the NCDs policy process [47]. Similarly, a wider stakeholder engagement including an NCD alliance network and other CBOs, which may include HTN patients, participated in the policy processes in Kenya [24].

Additionally, Pollock et al. [49] asserted that for the end effect of a policy, those who are to be affected by the policy must have the right to contribute to its formulation. Furthermore, the policies through their objectives addressed a range of issues concerning citizens’ health and wellbeing with strategies to optimise and improve quality of life. All the policy objectives were qualitative and may be problematic to measure, leading to poor monitoring and evaluation, poor verification of policy effectiveness, and plausible missed tracking of progress. This finding is supported by a similar policy review in Kenya [24] and other analogous policy reviews in Ghana [30]. Further, human resources were highlighted in all policies to assist policy actions with further plans to strengthen the training and engagement of adequate human resources, but minimal financial resources were committed. Tackling HTN control must begin at the primary health care (PHC) level, which is the entry point for access into the health services within Ghana [50,51,52]. This has been stressed by the WHO for the control and management of NCDs [52]. However, human resources for adequate implementation of most of these policies are lacking at the PHC level, thereby affecting the prevention and control of HTN [44,53]. Only a few of the policies had earmarked funds or indicated well-thought-out financial re-engineering to cater for policy initiatives. None of the policies had quantified and allocated resources to targeted activities. This situation is problematic and reveals the looming danger to the successful implementation of the policies with the inference that HTN control may be affected, posing a potential threat to reducing HTN control.

The importance of coordination and collaboration between public and private stakeholders was recognised by most of the policies. This is encouraging, as intersectoral collaborations in NCDs action and management have been cited in previous studies to be a challenge in Africa, especially when it comes to implementation [54]. All the policies included M&E plans. However, most lacked a detailed, methodologically structured, and objectively tailored M&E plan. This observation may not encourage donor agencies and partners to release funding since accountability processes are not transparent and tractable.

Population-level interventions for preventing HTN were also acknowledged across the policies as essential in controlling HTN in Ghana. All eight policies had significant interventions targeting environmental dimension of HTN control (i.e., changes in lifestyle behaviours such as adopting healthy eating, engaging in physical activity, rest, lower intake of sodium, monitoring of BP), food safety and quality, and use of essential medicines. The implication of these findings is that Ghana may be on track to achieving HTN control if these policies are religiously implemented. For instance, the provision of health insurance coverage for the population and the exemption of premium payments for the aged and children under 18 years of age is a social coverage that has significant health implication. Further, the government exemption of selected essential medicines from value-added tax and other forms of taxation would make the procurement, availability of essential medicines and equipment, and utilisation of health services more reasonably facilitate the control/reduction of HTN [41]. Studies have shown that people with health insurance have high healthcare utilisation compared to those without health insurance cover [55]. The frequent use of health services may lead to early detection of risk factors and control of HTN. In addition, previous studies in Ghana [56] and other places [57] have shown a positive association between health insurance coverage and HTN control. The other environmental-level interventions such as healthy dietary choices and regular physical activity both critically prevent and control secondary and primary HTN [25,58].

Few of the policies had additional consideration for the detection of HTN, whereas the majority did not consider any concrete plans for HTN risk detection. This finding mirrors similar review findings where few policies integrated HTN detection, treatment, and control within the health services. These findings raise significant concerns about Ghana’s progress towards HTN control/reduction, as the race to control the HTN epidemic cannot be adequately handled by one or two institutions. Even though multisectoral action has been observed in the policy formulation processes, similar to other findings across Africa [44,54], policies integration has been significantly lost even when all the policies originated from the same ministry of health. It is suggested that future policy development should reflect the “one-health approach” and integrate the policies to derive maximum impact.

Additionally, two of the policies recognised the need to adopt evidence-based guidelines to manage and control HTN [25,41]. This implies that priority for HTN control is still under-recognised as a major health emergency that needs concerted efforts to control. This situation could lead to poor awareness and knowledge on evidence-base guidelines for the treatment and control of HTN at both the health professional level and patient level, as health policy documents, which must be advocates of evidence-based guidelines for disease control, appeared not to have been captured in national policy documents. It is also plausible that because evidence for disease control keeps evolving, capturing the evidence-based guidelines at the time of the policy formulation in a policy document that may take some years to be amended may pose significant conflict when new evidence becomes available.

Task-sharing/shifting is another evidence advocated for in the management and control of HTN in Ghana and, by extension, Africa [23]. This is due to the dearth in specialist workforce in the management of diseases such as HTN. Only one of the policies reviewed captured a plan on applying task-shifting as a stop-gap measure [25] (p. 30). However, randomised control trials and other cost–benefit analyses have provided evidence that task-shifting is beneficial in HTN control in Ghana [56,59,60] and elsewhere [61]. This situation may likely shift preventive mechanisms to curative measures, especially at the PHC level, which serves as the main entry point for care, but mostly, the deficit is in specialist health personnel. The task-shifting strategy entails the appropriate transfer of primary care responsibilities from physicians to non-physician health care professionals, easing physician workload [52]. The doctor-to-population ratio in Ghana is 1:7374, which is much higher than the WHO-recommended number of 1:1000 [16]. Employing task-shifting, other health workers would be able to manage and treat non-complicated instances of HTN, while high-risk patients would be referred to the specialist [59]. Besides, the WHO and others have recommended task sharing within the health sector for the care of chronic conditions such as HTN to offset the impact of a dearth of highly qualified professionals [61,62]. It is thus recommended that policy makers and NCDs coordinators provide periodic in-service training for all low-cadre staff to constantly update them on current guidelines in the treatment and management protocol of HTN.

The finding on research evidence to guide interventions was implicit in three [25,37,41] out of the eight policies reviewed. These polices stipulated anchored plans to collaborate with research institutions and academia for research interventions. This is consistent with a previous study that reported a low research involvement in NCDs management [63]. Low research in NCDs such as HTN has been noted by the GHS, who integrated NCDs research into its national research agenda policy and advocated for a West-African-led NCDs research agenda to help provide an antidote to the rising NCDs prevalence in West Africa [63]. Even though these are steps in the right direction, it is recommended that future revisions of these national health policies integrate HTN research interventions to collaboratively tackle HTN control/reduction in Ghana. In addition, specific research interventions on HTN, which is a major risk factor for most CVDs and NCDs, should be targeted, as evidence shows that HTN control has a beneficial effect on the control of so many NCDs [64,65].

The review further uncovered three policies with provisions for universal coverage on detection, treatment, and control of HTN [25,37,38] and for ensuring the availability of essential equipment and medicines for managing HTN. Although a challenge, these findings give a glimmer of hope for reducing/controlling HTN by 25% by the year 2025. Generally, the interventions from the included policies present a narrower perspective and affirm the findings of Boudreaux et al. [66] on a review of NCDs policies in SSA. Given that there is low adherence to the PASCAR’s action plan in the policies, there is a need for a review of current policies. Additionally, it will be important to examine the experiences of stakeholders such as patients, health professionals, and policy makers to bridge the gaps and ensure that future policies are effectively implemented.

### Strength and Limitations

The study’s relevance and strengths are bolstered using a validated model to analyse policies/reports on Ghana’s progress towards HTN control by 2025 using an established evidence-based ten-point action plan for HTN control in Africa. This review is the first of its kind to track Ghana’s progress towards reducing HTN by 25% by 2025 using the PASCAR 10-point action plan for HTN control. To find as many policies and grey literature as feasible, the review explored a variety of sources, including official government websites, ministry websites, and *Google Scholar*. The review, however, was limited to Ghana and policies/reports that were available online and in English. The policies may not fully represent all procedures and actions involved in HTN policymaking process and HTN control/reduction in Ghana. Again, the policies may not fully capture all the procedures and actions required in developing policies to reduce HTN. The scoping review is further limited by its dependence on secondary data gathered only from documents. Gathering primary data from those who established the policies and initiatives as well as policy implementers and diverse beneficiaries such as HTN patients and networks would have improved the findings. This, however, does not invalidate the study findings.

## 5. Conclusions

Based on the 10-point action plan for HTN control in Africa, this study reviewed Ghana’s healthcare policies and grey literature using a validated technique to assess the country’s progress towards HTN control by 2025. The current policy initiatives in Ghana have low adherence to PASCAR’s evidence-based plans for hypertension control/reduction. The low HTN integration measures across most of the policies, with poor financial allocations to implement control/reduction measures, may stifle efforts at improving the country’s HTN situation to achieve global reduction/control targets by 2025. Moreover, the low task-sharing and lack of involvement of HTN patients in health policy formulation and implementation processes casts doubt on Ghana’s ability to adequately meet global HTN control/reduction targets by 2025. This review invites health policy makers and stakeholders within health systems to re-evaluate the current health sector policy strategies towards HTN control/reduction. The methodology employed in this review could be applied to other sector policies in Ghana and Africa to evaluate adherence to policy issues.

## Figures and Tables

**Figure 1 ijerph-20-01425-f001:**
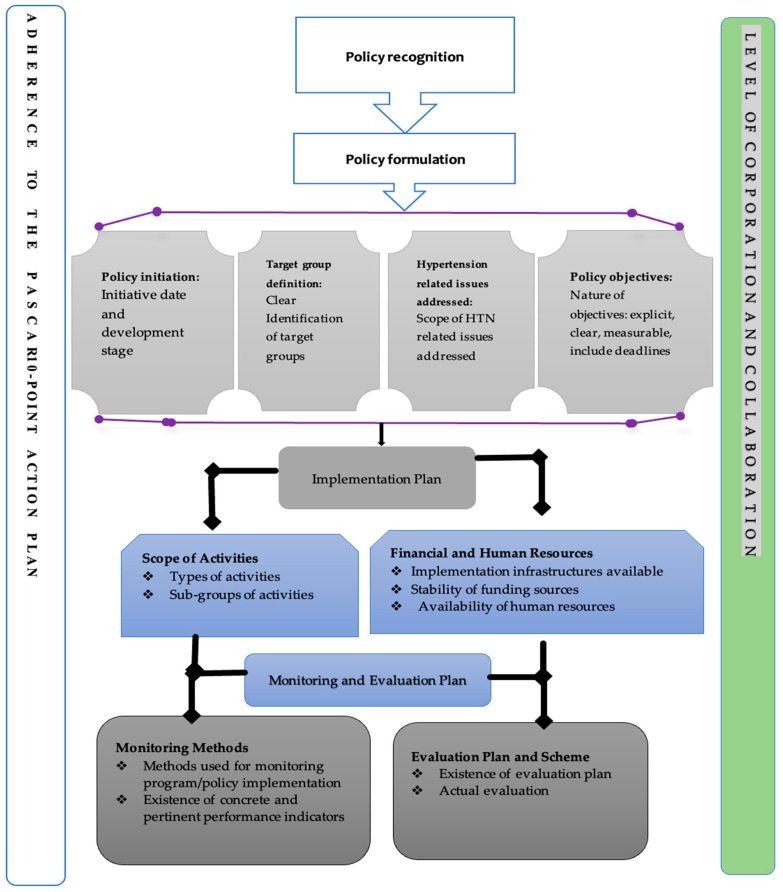
Adapted conceptual framework for evaluating programme and policy design. Source: Calves [33].

**Figure 2 ijerph-20-01425-f002:**
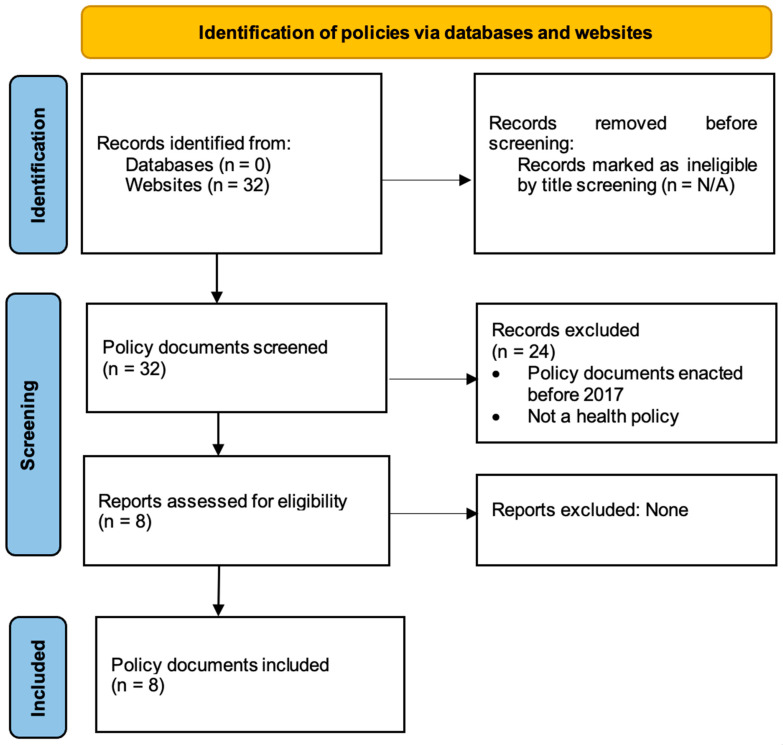
PRISMA flowchart.

**Table 1 ijerph-20-01425-t001:** Characteristics of health policies developed from 2017 to 2022 in Ghana.

	Policy/Report Title	Political Recognition	Issues Addressed Concerning HTN	Target Group Definition	Composition of the People Who Developed It	Policy Objectives (Quantitative or Qualitative)	Scope of Activities or Strategies Targeting HTN	Resources: Financial and Human	M & E Plan/Performance Indicators	Cooperation and Collaboration (Number of Partners Involved)	Involvement of Persons with HTN	Accessible Online?	Mention of HTN
1	National health policy (2020)	√	√	General population	Multidisciplinary group without inputs from HTN association or group	Qualitative	√	√	√	√	X	√	√
2	Ghana’s roadmap for attaining universal health coverage 2020–2030 (2020)	√	√	General population	Multidisciplinary group without inputs from HTN association or group	Qualitative	√	√	X	√	X	√	√
3	National policy: non-communicable diseases (2022)	√	√	People with non-communicable diseases	Multidisciplinary group including members of HTN Association	Qualitative	√	√	√	√	√	√	√
4	Health commodity supply chain master plan (2021)	√	√	General population	Multidisciplinary group without inputs from HTN association or group	Qualitative	X	√	√	√	X	√	X
5	Health Sector Medium Term Development Plan (2021)	√	√	General population	Multidisciplinary group without inputs from HTN association or group	Qualitative	√	√	√	√	X	√	√
6	National Medicines Policy (2017)	√	√	General population	Multidisciplinary group without inputs from HTN association or group	Qualitative	X	√	√	√	X	√	X
7	National food safety policy (2022)	√	X	General population	Multidisciplinary group without inputs from HTN association or group	Qualitative	X	√	√	√	X	√	X
8	Ministry of health client service charter (2020)	√	√	General population	Multidisciplinary group without inputs from HTN association or group	Qualitative	X	√	X	√	X	√	X

√, described; X, not described; HTN, hypertension.

**Table 2 ijerph-20-01425-t002:** Adherence to PASCAR’s 10-point action plan for hypertension control.

Policy/Report Title	Population-Level Interventions for Preventing Hypertension	Additional Plan for the Detection of Hypertension	Funding and Resources for the Early Detection, Efficient Treatment, and Control of Hypertension	Adopt Simple and Practical Clinical Evidence-Based Hypertension Management Guidelines	Support High-Quality Research to Produce Evidence That will Guide Interventions	Annually Monitor and Report the Detection, Treatment and Control Rates of Hypertension, with a Clear Target of Improvement by 2025, Using the WHO STEP Wise Surveillance in All Countries	Integrate Hypertension Detection, Treatment, and Control within Existing Health Services	Promote a Task-Sharing Approach with Adequately Trained Community Health Workers	Ensure Availability of Essential Equipment and Medicines for Managing Hypertension at all Levels of Care	Provide Universal Access and Coverage for Detecting, Treating, and Controlling Hypertension
National health policy (2020)	√	X	X	X	√	√	X	X	√	√
2. Ghana’s roadmap for attaining universal health coverage 2020−2030 (2020)	√	√	X	X	√	√	X	X	√	X
3. National policy: non-communicable diseases (2022)	√	√	√	X	√	√	√	√	√	√
4. Health commodity supply chain master plan (2021)	√	X	X	X	X	X	X	X	X	X
5. Health Sector Medium Term Development Plan (2021)	√	√	√	X	X	X	X	X	√	X
6. National Medicines Policy (2021)	√	X	X	X	√	X	X	X	√	X
7. National food safety policy (2022)	X	X	X	X	√	X	X	X	X	X
8. Ministry of health client service charter (2020)	√	X	X	X	X	X	X	X	X	X

√, described; X, not described.

## Data Availability

Not applicable.

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
