# Peer review of "Ghana’s Adherence to PASCAR’s 10-Point Action Plan towards Hypertension Control: A Scoping Review"

_ijerph, 2023, doi:10.3390/ijerph20021425_

Round 1

Reviewer 1 Report (Previous Reviewer 2)

Authors have addressed comments.  Nice contribution. 

This manuscript is a resubmission of an earlier submission. The following is a list of the peer review reports and author responses from that submission.

Round 1

Reviewer 1 Report

The authors have worked on an important topic in relation to NCD management in Ghana. Health policies could mean little in practice if they stay only on paper. A few comments -

(1) Review language in some parts of the manuscript, including minor brushing up for grammar and sentence structure. 

(2) A quicks summary of the PASCAR HTN roadmap at an early stage of the manuscript will help readers to understand the context much better before going into the detailed benchmarking in Table 2

(3) It is slightly surprising that it was discovered only at the last stage of screening that there were 4 reports that did not contain any strategy or interventions for managing HTN - does it mean refinement would have been necessary in the search terms (e.g. "strategic plan of Ghana"), or the screening process?

(4) While the manuscript focuses on Ghana, a brief summary of/ comparison with relevant policies in a few countries of socio-economic development similar to Ghana would put the whole analysis in much better context. 

Reviewer 2 Report

The idea of reviewing policies according to a defined set of standards that encompass the policy process from beginning to end is an excellent one.  I think that the approach used in this paper and even the particular policies examined could the basis of a very interesting paper.  However, I had some major concerns about this paper as currently constituted:

1) While I understand the importance of hypertension, it seems odd to policies that were intended to be more general in nature against the standard of their focus on hypertension.  This feels arbitrary. 

2) Some of the assessments also seem arbitrary. For example, on line 357, the authors claim that "promote healthy eating" is an explicit strategy.  That would hardly seem to be the case.  

3) At the end of the day, as the last point indicates, these are very broad policies.  So much of the meat will lie with regulation and implementation that it's truly hard to tell what their impact will be. 

That said, the policies form the basis for that regulation and implementation.  Therefore, identifying areas of weakness before we even look at regulation and implementation does make sense.  So, for example, the authors' point about the lack of patient involvement in the development of these policies is a good one.  Issues of workforce and other resource sufficiency are also important.

I suggest writing a different paper that uses the data collected but looks at it from a broader lens, e.g., NCD control, and is clearer about the criteria for assessment.